Research

# 'Because my brain isn't as active as it should be, my eyes don't always see': a qualitative exploration of the stress process for those living with posterior cortical atrophy

Emma Harding,[1] Mary Pat Sullivan,[1,2] Rachel Woodbridge,[3] Keir X X Yong,[1] Anne McIntyre,[3] Mary L Gilhooly,[3] Kenneth J Gilhooly,[3] Sebastian J Crutch[1]

¹Dementia Research Centre, University College London, London, UK
²Faculty of Applied and Professional Studies, Nipissing University, North Bay, Ontario, Canada
³Department of Clinical Sciences, Brunel University, Uxbridge, UK

**Correspondence to**
Emma Harding;
emma.harding@ucl.ac.uk

## ABSTRACT

**Objectives** To explore the stress process for individuals living with posterior cortical atrophy (PCA) and their families.

**Design** A qualitative study using in-depth semi-structured dyadic and individual interviews with people living with a diagnosis of PCA and a family carer. Interview transcripts were thematically analysed.

**Setting** Participants' homes.

**Participants** 20 individuals in the mild to moderate stages of PCA and 20 family carers.

**Findings** Three major themes were identified: (1) the diagnostic journey: mostly an unsettling and convoluted process, owing to the early age of onset, rarity and atypical symptom profile of PCA. (2) Interactions with the physical environment: profound difficulties with functional and leisure activities were usually compensated for with adaptations maximising familiarity or simplicity. (3) Implications within the psychosocial environment: symptoms impacted individuals' sense of independence and identity and required reallocations of roles and responsibilities. Ongoing uncertainties and the progressive nature of PCA caused most dyads to take a 'one day at a time' approach to coping. Relatively well-preserved insight and memory were a benefit and burden, as individuals shared the illness experience with family members and also compared their current situation to pre-diagnosis. The experience was framed by background and contextual factors and understood within an ever-changing temporal context.

**Conclusion** The stress process in PCA is characterised by uncertainty and unpredictability from diagnosis through to ongoing management. The provision of tailored information about cortical visual problems and associated functional difficulties, time-sensitive environmental adaptations to help those with PCA to identify what and where things are and psychosocial interventions for the marital/family unit as a whole would be useful to improve both functional status and psychological well-being. Future research exploring (1) stress and coping in the later stages of PCA and (2) the nature and impact of visual impairment(s) in typical Alzheimer's disease would be worthwhile.

### Strengths and limitations of this study

► As the first qualitative study of those living with posterior cortical atrophy (PCA), this paper provides original, in-depth insights into the subjective experiences of those with dementia-related visual impairment.

► Conducting both individual and dyadic interviews within participants' homes permitted both the multiple perspectives of people with the diagnosis and their family carer to emerge and a rich understanding of the physical and psychosocial context within which daily difficulties owing to visual processing problems were experienced.

► As well as providing empirical description of the illness experience, in using a conceptual framework (the Stress Process Model), this study also makes a broader contribution to social science and the field of dementia research.

► Interviews were conducted at one time point, but owing to the progressive nature of the disease, future longitudinal research would be valuable to develop the current study findings.

► The interview method relies on participants' abilities to accurately recall their own experiences and while those with PCA, especially in the earlier stages, can have relatively well-preserved memory, disease severity varied across the current sample, and a subgroup of those interviewed were demonstrating some memory impairment during interview.

## BACKGROUND

There are an estimated 850 000 people currently living with dementia in the UK,[1] and it is estimated that 5% of these (approximately 42 500) are cases of young-onset dementias, with symptoms beginning before the age of 65 years.[2] Posterior cortical atrophy (PCA), originally called Benson's disease, is a rare form of dementia that is typically early in onset with symptoms usually

beginning between the ages of 50 years and 65 years.[3] The underlying pathology in the majority of individuals is Alzheimer's disease (AD), although a small number of cases attributable to Lewy body disease and corticobasal degeneration have been reported.[4 5] The prevalence is unknown. PCA is characterised by initial neurodegeneration towards the back of the brain, specifically in the parietal, occipital and occipitotemporal regions.[6] Correspondingly, the initial symptoms predominantly relate to cortical visual impairment, particularly deficits in visuospatial and visuoperceptual processing. Other characteristic symptoms relate to impairments associated with posterior functions, including literacy, spelling and numeracy.[6 7] While current clinical criteria cite visual processing impairments with proportionally less impaired memory as core diagnostic features of PCA, patients may in fact exhibit memory impairments at initial presentation.[4 8 9] When compared with other dementias, PCA is relatively under-researched. The majority of research into PCA is concentrated around establishing the neuropsychological, cognitive and imaging profile(s) of those with the diagnosis.[4 9–11] Anecdotal and laboratory-based evidence suggests ways in which PCA might impact on people's daily lives, including problems with reading, driving and localising objects in space,[9 12–14] but there is a paucity of research focusing on the everyday impact of living with the diagnosis for individuals and families.

One recent paper investigated the impact of PCA on activities of daily living, documenting difficulties with performance of everyday skills for people with PCA (including operation of appliances, writing and handling money) and self-care (including dressing, feeding and bathing) compared with predominant deficits in memory, motivation and orientation of people with typical, amnestic presentations of AD (tAD).[15] In another recent paper, Suárez-González et al[16] investigated the neuropsychiatric profile of people with PCA. They found similar increases in depression, apathy, irritability and anxiety for those with PCA to participants with tAD.

In the absence of disease-modifying therapies for AD and other forms of dementia, environmental and psychosocial interventions to improve the quality of life and well-being of those living with it hold particular significance.[17–19] Better understanding the needs of those with different rare and/or young-onset dementias will be an important step in developing effective environmental or psychosocial interventions. A second generation of literature is beginning to delineate the particular experiences of those with less common forms of dementias such as young-onset dementias[20 21] and those with atypical symptom profiles like behavioural variant frontotemporal dementia.[22] Developing a greater understanding of the day-to-day impact of dementia in relation to visual problems will be a timely addition to this, not least because those with tAD may also go on to have cortical visual impairment, likely later on in their diagnosis and as such at a time when they may not be so able to articulate their experiences of the symptoms.[23]

These varied everyday impacts of dementias that pose challenges for quality of life and well-being are often considered within a broad conceptual category of stressors, defined as demands that are considered to exceed a person's available resources. Cognitive, emotional and/or behavioural attempts to manage these demands are often approached and studied as coping strategies. Much attention has consistently been given over the past few decades to understanding what contributes to informal caregiver stress and what facilitates coping, given the huge societal contribution informal carers make by continuing to care for loved ones with dementia at home.[24 25] The stress-coping approach is widely acknowledged as dominant within this literature,[26 27] and its popularity is exemplified by the numerous reviews into stress and coping in dementia noted by Gilhooly et al.[19] There are multitudes of studies looking at the chronic and particular psychological, socioemotional and practical stressors and strains faced by carers of people with dementia[28 29] and also the nature and efficacy of the many practical, emotional, psychological and social coping strategies employed to mediate this.[30 31]

The current study sought to maximise on the relative abilities of those with PCA to reflect on and communicate their experiences. Using the Stress Process Model[32 33] as a conceptual framework, we present findings from a qualitative exploration of the stresses associated with mild to moderate stage PCA and responses to these over time. The Stress Process Model outlines primary stressors that result directly from the disease itself, secondary strains that may follow and both internal and external factors that mediate both of these in shaping outcomes. Having been developed to conceptualise informal caregivers' experiences,[32] it has since been adapted for individuals with dementia,[33] and in doing so acknowledges the multiple perspectives to be taken into account when understanding the dementia experience. More specifically, the study aimed to explore the potential of the physical environment to contribute to and/or mediate the stress process owing to the prominent visuospatial and visuoperceptual symptoms.

## METHODS
### Design/sampling
A qualitative design was deemed appropriate in order to gain rich detailed accounts and for possibly unanticipated insights to emerge.[34] This was considered important because of both the paucity of knowledge about the impact of PCA on individuals' subjective experiences and also the distinctiveness of the symptom profile. In-depth semi-structured interviews were chosen owing to the abilities of those with PCA to recount their experiences and, again, to facilitate the collection of data of sufficient richness and depth. Participants were recruited via the Specialist Cognitive Disorders Clinic at the National Hospital for Neurology and Neurosurgery, University College London Hospitals NHS Foundation Trust. Inclusion criteria were a

**Table 1** Five verification strategies for attaining reliability and validity in qualitative research[39]

| Verification strategy | Explanation | How or where demonstrated in the current study |
|---|---|---|
| Methodological coherence | Ensuring congruence between research question and methods | Background (Stress Process Model) Methods (design/sampling, ie, community-based sample; data collection, ie, individual and dyadic interviews in order to gain both shared and individual perspectives; data analysis, ie, qualitative approach for study of little-known topic) |
| Appropriate sampling | Participants who best represent or have knowledge of the research topic | Methods (design/sampling, ie, community-based sample; data collection, ie, individual and dyadic interviews in order to gain both shared and individual perspectives; broad range of disease severity) |
| Collecting and analysing data concurrently | Establishing an iterative interaction between what is known and what one needs to know | Methods: moving between data collection and data analysis, including: memo-writing (keeping an ongoing log of analytical thoughts and ideas) amendments to interview schedule (adding questions/prompts to further explore emerging areas of interest, eg, role changes) field notes (written together by the two interviewing authors—MPS and RW or RW and EH—immediately after the interview to document initial responses and reflections on the data collected) |
| Thinking theoretically | Constant, cyclical process of checking that emerging ideas are reconfirmed in new data | Methods: moving between data collection and data analysis, including: memo-writing (keeping an ongoing log of analytical thoughts and ideas) amendments to interview schedule (adding questions/prompts to further explore emerging areas of interest, eg, role changes) field notes (written together by the two interviewing authors—MPS and RW or RW and EH—immediately after the interview to document initial responses and reflections on the data collected) |
| Theory development | Moving between microperspective to macroconceptual/theoretical understanding | Results (major and subthemes, supporting quotes and explanatory commentary) Discussion (compatibility with existing literature, eg, empirical—relationship impact and theoretical—utility of the Stress Process Model; research and clinical practice implications; suggestions for future work, eg, other rare dementia populations) |

confirmed diagnosis of PCA and an accompanying family member or familiar other also willing to participate.

Twenty individuals with PCA (12 female; 8 male) took part in the interviews, and the mean age was 68 years (7.66 SD). Scores on the Mini Mental State Examination ranged from 10 to 29 (mean=20.05; SD=6.54), indicating mild to moderate dementia. Twenty spouses/family carers took part (10 female; 10 male); in 18 cases, this was a spouse, in one case the dyadic relationship was mother–daughter and in the other case it was aunt–niece. One spouse (female) opted not to take part in an individual interview but did participate in the dyadic one. Only one participant lived alone. The number of years since diagnosis ranged from 0 to 12 (mean=3.31; SD=2.75), and the number of years since subjective onset ranged from 2 to 14 (mean=6.39; SD=3.26).

A comparative sample of people living with tAD (n=17) and their family carers (n=17) were also interviewed and findings from the subsequent analysis of that data will be reported in another paper.

### Ethical approval
Informed consent was obtained from all participants. After the interviews, the researchers conducted a short debrief providing further information and contact details in case of any issues or causes for concern related to the study.

### Data collection
Individual and dyadic interviews were conducted at participants' homes (by EH and RW or MPS and RW). Dyads were interviewed together and then separately in order to capture the dyads' shared experience,[35] to allow the family carer to supplement the person with PCA's account in the case of additional, secondary memory impairment and to provide the opportunity for individuals to provide information they might not feel comfortable to disclose in the presence of their family member.[36] The interview schedule covered contextual factors (personal, marital and occupational history and current family situation), the diagnostic journey and daily difficulties and coping strategies within the home environment. In total, interviews lasted between 3 hours and 4 hours per dyad. The home visit also involved a walk-around[37] of areas of the home posing particular challenges to participants or where they had implemented assistive strategies.

Audio-recorded interview files were transcribed, and a random portion was checked for quality. All names and place names were changed.

**Table 2** Neuropsychological scores of patients with PCA (n=20)

| Test | Max. score | Raw score (mean±SD; *range*) | N <5th percentile cut-off | Comments |
|---|---|---|---|---|
| Short Recognition Memory Test* for words (memory)[75] | 25 | 17.98±4.857 12–25 | 10 | ~<5th percentile (cut-off 19) |
| Concrete synonyms test (language)[76] | 25 | 21.33±3.105 12–25 | 7 | 25th–50th percentile (cut-off 18) |
| Fragmented letters (VOSP) (visuoperceptual)[77] | 20 | 1.60±2.703 0–10 | 19 | ~<5th percentile (cut-off 16) |
| Dot counting (VOSP) (visuospatial)[77] | 10 | 2.15±2.961 0–11 | 18 | ~<5th percentile (cut-off 8) |

*Behavioural screening tests supportive of PCA diagnosis.
PCA, posterior cortical atrophy; VOSP, visual object and space perception battery.

## Data analysis

Interview transcripts were uploaded into Atlas.ti qualitative data analysis software (V.7). The data were analysed using thematic analysis,[38] which was selected owing to its flexibility and accessibility. These were considered important factors given that this is the first qualitative exploration of the experiences of individuals with PCA and also the wide range of health professionals who may find relevance in the findings. Two members of the research team (EH and MPS) were responsible for the analysis. They first familiarised themselves with the data with multiple read-throughs of transcripts before creating an initial coding framework based on existing literature on stress and coping in dementia, the study research questions and the initial familiarisation process. This coding framework was flexible, and new codes were added as required, following discussion and agreement. Each dyads' set of three interviews (person with PCA, family carer and joint) constituted one case for analysis, and the cases were divided among the two authors EH and MPS. Once all 60 transcripts had been analysed and assigned initial codes, the codes were sorted into broader themes. Some codes were organised into major themes, some into subthemes and others became theme headings themselves. Themes and coded extracts within them were then reviewed in terms of their relevance, distinction from each other and coherence, and codes were reallocated or reorganised where required. Themes were then defined and named in such a way that they offered a coherent and consistent account of the data.

## Quality assurance

Ongoing discussions acknowledge the complexity but necessity of assuring the quality of qualitative research. We draw on a notion of rigour in qualitative research suggested by Morse *et al*.[39] Morse *et al*[39] identified five verification strategies for attaining reliability and validity in qualitative research that require consideration by the researchers throughout the research process, as opposed to criteria for reliability and validity that are determined post hoc and only by readers. Table 1 outlines these five verification strategies and where or how they are or were addressed in the design, conductance and write up of this study.

## Member checking

Member checking is the process of presenting qualitative research findings to respondents or participants and inviting their feedback and/or checking for resonance of the findings with their own experiences.[39] The advantages and disadvantages of member checking are much contested,[40 41] and details of the debate are beyond the scope of this paper, but several issues bear particular relevance here. First, there is a concern that participants may inaccurately recall their original account, and this is perhaps more likely in the case of progressive cognitive decline. Second, developments since the participants gave their original accounts may have changed their perceptions or how they may now choose to respond to the same questions and this too may be of increased likelihood for those living with a progressive condition. For these potentially confounding reasons, we sought external validation of the current findings via two regional PCA support groups, made up of people with a diagnosis of PCA and their family carers (as in the study sample). Study findings were presented to both groups and comments invited, and there was a general consensus across both groups that the themes elicited here were compatible with support group members' own experiences. Many support group members went on to share their experiences by way of demonstrating the overlaps and coherence with the results of the current study.

To further establish the quality of the current research project, the project was conducted in accordance with the COnsolidated criteria for REporting Qualitative research criteria (see online supplementary appendix 1).[42]

## Findings
### Neuropsychological assessment

All participants with a diagnosis of PCA had previously (within 6 months) completed a selection of neuropsychological tests of memory, language and visual processing skills (visuoperceptual and visuospatial). Descriptive data relative to normative data sets appropriate for the

mean age of the group are presented in table 2. Mean participant scores on the Short Recognition Memory Test (words), fragmented letters (visuoperceptual) and dot counting (visuospatial) tasks were below the 5th percentile, with mean scores on the concrete synonyms test falling within a normal range. About half of participants' individual scores on the tests of memory (Short Recognition Memory Test for words) and language (concrete synonyms test) fell below the 5th percentile (n=11 and n=9, respectively), whereas almost all participants' individual scores on the tests of visuoperceptual (fragmented letters) and visuospatial (dot counting) processing skills fell below the 5th percentile (n=19 and n=18, respectively).

## Qualitative interviews

Participants described a range of ways in which the diagnosis of PCA and the associated symptoms contributed to stress over time and various strategies they and close others employed in response. The findings comprise three central themes that highlight some of the diagnosis-specific characteristics of PCA: the journey to diagnosis; interacting with the physical environment including managing activities of daily living, navigating the outside world and use of aids and adaptations; and implications for the psychosocial environment such as maintaining independence and the adoption of a one day at a time approach to coping. Within these key themes, there were associated temporal variations due to transitions associated with stage of life and others more representative of living with a neurodegenerative illness. The illness experience was framed by numerous contextual and background factors (eg, existing relationship quality, personality factors (eg, being a gregarious/cautious/easygoing person) and life stage transitions).

## The journey to diagnosis

Initial symptoms were often described as incongruous or hard to pinpoint but nevertheless were an indication that something was wrong. Often, struggles with very familiar activities were first noticed:

> Everything was hard for about a year, and [I was] beginning to feel there's something not right here, because I couldn't work out sort of basic things. (Participant with PCA)

It was not uncommon for individuals to describe a problem that arose in their workplace, for example, reading financial accounts, or difficulties judging distance while driving. These challenges were often attributed to the health of their eyes and presented in stark contrast to the typical short-term memory problems first noticed in cases of tAD.[43]

With the everyday nature of tasks becoming difficult, this was inherently unsettling and a primary source of stress in and of itself. The majority (n=19) of individuals consulted eye health professionals (eg, optometrists) in the first instance and underwent various inconclusive eye health tests despite the cortical nature of their visual impairment. With hindsight, couples reflected on how the stress had been exacerbated and drawn out because of the lack of knowledge of the illness among the healthcare professionals they were consulting. For some (n=10), this was complicated by concurrent eye health issues, further delaying diagnosis. Experiences with general practitioners (GPs) (n=17) were similarly reported to be frustrating because of a lack of answers or appropriate and timely onwards referrals. For some, even a referral to a neurologist did not guarantee a diagnosis:

> That would have saved me a lot of trouble if I'd believed myself. And it took ages, we went through about five or six neurologists… Nothing. It was just dreadful because I kept thinking if I tell them what's wrong, what's happening and the symptoms, they're bound to know… And nobody knew. (Participant with PCA)

A minority of participants (n=3) reported a timely and efficient diagnostic process, and this was mostly attributed to their own efforts towards information seeking but for one male was attributed to the 'pot-luck' of his GP's professional connections. The remaining dyads (n=17) described stress caused by having to persist in their search for a diagnosis for what they considered to be an unacceptably long period of time. This is consistent with existing literature that describes the benefits of and need to prioritise the early diagnosis of dementia for individuals, families and society[44 45]; however, this study highlights the particular barriers those with PCA face in receiving a timely diagnosis, owing to the rarity of the condition, associated lack of professional awareness and atypical symptom profile that lead them to exploring an eye health route. Consistent with this idea of the importance of having knowledge of the illness, many participants explicitly stated the relief they experienced when the diagnosis was provided (n=11).

Following diagnosis, there was a widely reported lack of accessible information. This has previously also been reported by those adjusting to a diagnosis of tAD[46 47] and young-onset dementia specifically,[48] and the current findings therefore add to knowledge about the varied types of dementia for which advice and information are needed. The rarity of PCA had the potential to be an ongoing source of stress over time in that those living with the diagnosis repeatedly found themselves better informed on the condition than healthcare professionals they came into contact with, often having to re-explain the syndrome and their symptoms on multiple occasions.

## Interacting with the physical environment

The nature of the symptoms (ie, predominantly visual) meant effectively interacting with the physical environment was the predominant issue. This included interactions within and outside the home environment and with activities both functional and 'fun'.

Every participant described complications with completing self-care tasks, most prominently dressing and cooking. Difficulties with dressing included finding

or selecting clothes or shoes, orienting them and using fastenings:

> I do struggle a bit sometimes in working out which way round shirts go… If it's all in a big heap, which it generally is, it's just a question of I will perhaps turn it round, sort of, two or three times before I work out where the collar is. (Participant with PCA)

Dressing problems were exacerbated by distinguishing clean versus soiled clothes, seeing the closet and bedrooms being a shared space. Dressing assistance was frequently obtained from a family member or by decluttering, organising and simplifying the bedroom environment. In contrast to those with tAD, where problems with dressing may be attributable to problems with sequential task performance and attention,[49] most of those with PCA articulately described clear visuopercpetual and visuospatial processing problems underpinning their difficulties. Those with PCA were able to be similarly articulate about their choices and preferences around clothing and remained motivated to initiate dressing activities, which may be in contrast to those with tAD whose dressing may more commonly be disrupted by temporal disorientation and lack of motivation.[50]

With cooking, typical problems were with locating ingredients/equipment in the kitchen, reading labels, following recipes, using appliances or confidently and safely handling hot materials. For cooking or other household tasks, some individuals attempted to use visually salient strategies such as labelling cupboards or putting a red dot on the start button of an appliance. The effectiveness of these strategies varied and more typically individuals retreated from these activities. As with dressing, when these difficulties are reported in the literature on tAD, they are usually attributable to declining executive function skills.[51]

Of particular importance to participants were difficulties they had in engaging with a wide range of hobbies and interests including reading, DIY, sports and arts activities. The impact of this seemed heightened by the typically young age of onset that saw all participants either approaching retirement age, having recently retired or reducing their working hours, and therefore allocating and looking forward to increasing time related to leisure activities:

> I think we thought we'd be going out to theatre and travelling, and things more, whereas I'm planning in 2015 to make it the year I'm going to go to the matinees, try that. But, travelling has virtually stopped. We were going to take John's mother to the Christmas markets, thinking, well, with her help, I can probably get John on and off the train, but it… with her breaking her hip, that's another thing, another holiday went because of that, and so we thought, this is the time, this will be the time in our life that we need to travel, and this is the time in life, for one reason or another, we can't… Yes, I think we thought this would be the golden years. (Family carer)

The stress at having to retreat from or renegotiate hobbies is consistent with a study that looked at aspects that are important for quality of life but challenged by dementia.[52] However, the barriers here were largely due to the specific visuospatial and visuoperceptual deficits, which is in contrast to a study by Giebel et al[53] in which carers described those with tAD as having difficulties with the initiation rather than performance of such activities.

Beyond the home environment, all participants had difficulties navigating the external environment either on foot, by car or on public transport. Particular challenges were wayfinding through crowds, reading signs or maps, general orientation and using stairs/escalators:

> I'm usually quite okay here but the minute I step outside the door it all goes mad… Life goes a bit strange, yes… even inside the village can be a bit strange but definitely when I go catching buses and dealing with… interacting with people, in general, no, it's not great, not great… I don't know what happens but it goes mad, yes. Not all the time and not every time but I'm much less comfortable and avoid, now, going unless I really need to go into town. (Participant with PCA)

Participants commonly stated they relied on routine responses to the environment and cues within it (eg, using the same underground line or bus) or environmental cues (eg, recognising a street by a particular shop or church), but naturally the external environment is not a stable one. In the case of (for example) a disruption in public transportation or a lorry blocking the view of an environmental cue, problems arose such as the person with PCA getting lost or disoriented. It is important to note that, for most, this was not due to forgetfulness, distractibility or other executive function deficits as might be expected in cases of tAD[54 55] but because of problems accurately perceiving visual information about the environment that would help individuals to work out where they were in relation to their target destination.

The difficulties that participants described in interacting with their physical environments are compatible with the sorts of neuropsychological deficits documented in people with PCA in the literature. Deficits like visual crowding, simultanagnosia, spatial navigational problems and apraxia[4 9 14] corroborate with the issues people described with locating and manipulating objects, reading and dressing in this study.

As described above, the tendency towards simplification and familiarity meant withdrawing from certain activities. In addition though, almost all participants described ongoing uncertainty and unpredictability associated with the disease profile, commenting that the difficulties were not reliably ever-present (n=16). This uncertainty was once again exacerbated by a reported lack of disease-specific provision and guidance. A minority of participants opted for off-the-shelf adaptations for those with eye health problems, for example, a symbol cane (n=3), or had been in touch with the

Royal National Institute for the Blind regarding visual aids (n=5) but most took a self-initiated, largely trial-and-error approach owing to the unusual, unpredictable and continually changing nature of the symptoms. This process was rife with uncertainty regarding if things would work, why they might not and over time, how long they would continue to work:

> Camilla can still read, so if it's just one word it's okay… but we've done different colour coding [on shampoo/conditioner bottles] and this sort of stuff but then she forgets which colour's which. So it's not, you know, it seems to be simple… but then there'll be some other obstacle along the way. (Family carer)

This family carer provides an example of the complexity of progressive cognitive decline where strategies to compensate for the dominant visual symptoms may rely on other cognitive capacities (eg, memory) that may also be affected to an extent. This was in contrast to those without memory impairment at the time of interview (n=9) who frequently relied on the familiarity of their environment to help them find their way or something they needed, often closing their eyes or feeling their way to minimise any confusing visual information.

Allen *et al*[56] recently described a similar trial-and-error approach to environmental adaptations being employed by community-dwelling people with tAD and their carers, but these were more often triggered by—and designed to ameliorate—difficulties associated with dominant memory problems (eg, using labels as reminders).

Acknowledging the temporal context, the participant above like many others referred to what his wife could 'still' do—appearing to demonstrate an anticipation of continuing decline over time. This is something commonly reported throughout the dementia literature in general.[57 58] In the case of PCA though, this was coupled with a relative paucity of accessible knowledge or professional guidance as to how the particular course of disease would progress.

### Implications within the psychosocial environment

Overall, there were broader psychosocial ramifications arising from these day-to-day difficulties such as maintaining independence and contemplating an uncertain short-term and long-term future.

The symptoms themselves and resulting difficulties in interactions within the physical environment naturally impacted individuals' ability to perform daily activities independently:

> I think that is the worst thing I can do nothing for myself so all the time you've got to ask somebody. (Participant with PCA)

Psychosocial strategies used to mediate the stress associated with performance of these daily activities were largely the provision of physical assistance and reallocation of responsibilities within the dyad or family. Over time, this often resulted in increasing feelings of dependence:

> One of my big problems is frustration that I can't do things I want to do. I don't need to do them but because I've always been able to do them it really irks me to have to phone my son up and say, come and put this together for me. (Participant with PCA)

This example illustrates a common phenomenon reported in the interviews—the way this perceived dependency resulted in changing roles and responsibilities within families. The extent to which this caused secondary strain for individuals seemed to be mediated by contextual factors including the age of onset, personality factors and previous relationship quality. Interestingly, the one couple who were recently married (<5 years) described their biggest challenges as being with organising the household responsibilities, though they were able to emotionally support each other well. This was perhaps owing to their less well-established household roles and responsibilities but their relatively recently established and continuing affection for each other.

With age of onset, individuals had often been at the peak of their careers and/or in a critical position within the family system (eg, looking after both children and elderly parents). Such activities and the roles they represented were often defining in terms of individuals' senses of self and identity:

> …he does all the ironing and everything now. And I say, no, that's my job… I'd always done it, you know. [He] was always working and I had the children and everything, and just did it… And I must admit, I sit here now and I think, I can't do anything. (Participant with PCA)

The threat that dementia and the associated decline in functioning can pose for a person's sense of identity and independence is well documented throughout the existing literature, even when functional capacity is disrupted by memory problems[58 59] rather than predominant cortical visual deficits as seen here.

Family carers (n=13) also described concerns they had about their competencies in taking on the new role of caregiver and reported the strain of uncertainties in knowing how and when to provide help. Many noted trying to strike a balance between getting things done, preventing their family member's distress or frustration and encouraging or facilitating their ongoing independence.

Any stress and strain for participants who had a diagnosis of PCA regarding the impact on identity, role and independence were arguably emphasised by their relatively intact long-term memory functions and abilities and inclination to reflect on and compare their previous experiences with their current situations. One way this caused stress was that the person with PCA could reflect on themselves and their declining abilities, and in doing so most (n=15) were concerned that they were becoming a burden to their partner or wider family. However, these relatively intact capacities were also able to contribute to relieving stress for the majority of couples who took

an interdependent or 'teamwork' (n=16) approach to managing the difficulties, via continued collaboration and joint problem solving. Those dyads who did not adopt such a team-based approach (n=4) appeared not to because of either factors relating to: their previous relationship quality (eg, living fairly independent lives); personality (eg, pride and stubbornness); or the cognitive decline of the person with PCA (eg, poor memory for shared events). The dominant 'teamwork' approach or sense of 'being on the same page' and navigating the illness experience together was made evident in shared story-telling during the joint interviews and in the corresponding and complimentary accounts given in the individual interviews. However, there were also some instances of discrepancies in the accounts of the person with PCA and their family carer. Sometimes differing recollections of events (eg, a family occasion, first symptom onset) would seem to be easily explained by the person with PCA's concurrent memory impairment, but in other cases, the discrepancies were harder to unpick. Several dyads offered differing accounts of the person with PCA's functional ability. In one case, a daughter attested that her mother was trying to not let us, the researchers, know how impaired she was by insisting she was still doing household chores and that the discrepancy could be explained as an exercise in self-presentation. In another case, a man with PCA and his wife disagreed over whether he could safely go for a walk unattended—she considered his wayfinding abilities to be too compromised and he considered her to be too closely monitoring him and disproportionately concerned. Day-to-day difficulties were also attributed to different underlying symptoms, for example, one man with PCA put his problem with navigating the stairwell at a relative's house down to the area being 'dim' and 'dark', whereas his wife put this down to him not remembering where he was correctly:

> …you still get lost on that landing…So it's a square area with four closed doors where does he go? And he never knows…like where is it, where on earth is it? So he can't retain the information…this is like every single time…he is walking down those steps for the first time. (Family carer)

These discrepancies—though not the common pattern in the current study—clearly highlight the importance of acknowledging potential differences in perspectives and the challenges these could pose in the day-to-day understanding of and responses to the symptom profile and its impact.

Despite these occasional discrepancies, many individuals with PCA (n=11) expressed extensive gratitude or feeling 'lucky' for their spouse/family carer while also normalising any symptom-specific dependency as one of many ways in which they and their family member worked together to manage life's challenges:

> If you get married, sickness and in health, you have to keep to these things… I'm old-fashioned enough to think… you know, if it were me, he would look after me, I have no doubt…So, you know, what I think, and this is what I say to him, we're married, we're two parts of a whole, so in many ways it affects me, because then when you are supposed to be as one, as a whole… then you have to look after the other half of you, and, you know, by keeping one half healthy, helps the other half. (Family carer)

There was a suggestion that a relatively preserved insight and ability to plan in people with PCA promoted a continuation of closeness and collaboration between dyads, potentially contrasting with previous studies involving participants with young-onset but memory-led dementias. Baikie[60] reported a loss of joint decision making in marital relationships, while separateness made up part of an overarching theme in a study by O'Shaughnessy et al[61] about the impact of dementia on the marital relationship. Similarly, Wright[62] described how a lack of awareness contributed to discrepancies in the accounts of people with dementia and their spouses regarding their experiences of tension within the relationship and also an overall reduction in shared meanings made about the illness experience. However, there are also reports of couples living with memory-led dementias taking a continued teamwork approach[46] and sustained reciprocity in consideration of the others' needs,[63] both of which serve as a useful reminder of our need to interpret with caution. An example of this within our own study was one of the male participants who demonstrated concurrent memory problems along with his dominant visual processing deficits, whose wife commented:

> Everything's more flat, yes. And he's got no sense of time, so whereas before… he would go…oh yes…this is the year that she said was our silver wedding anniversary…now he's got to rely on the children to say… we'd better do something about it…I found it painful. (Family carer)

Two points seem salient here. First, the dementias progress in different ways for different people, and second, given the variation, states of separateness or connectedness, or teamwork versus independence, do not apply discretely and exclusively to a group of people with one diagnosis and not to another. The participants with PCA interviewed here will likely progress to have more memory challenges over time, potentially posing additional challenges to shared meaning-making, and those in the early stages of more typical dementias are increasingly shown to be able to reflect reliably and accurately on their own experiences and abilities,[63 64] something essential for joint problem solving and shared decision making.

Another factor to consider is the sociocultural context within which research questions are framed and studies are carried out. As the value put on quality of life surpasses that of quantity and as cures for dementias remain elusive, attention has shifted to ideas of preserving personhood of

those with dementia and more recently still, the couple-hood of dyads living with a diagnosis. Acknowledging the person and not just their disease has also meant a shift in focus from seeking to document deficits, loss and failings and attempts to capture the whole breadth of experience as attempted here: the positives, strengths, closeness and resilience.[65 66]

## One day at a time

Beyond diagnosis and in terms of longer term ongoing coping with the diagnosis, uncertainty and a lack of knowledge persisted, this time with regards to what to expect and how the disease would progress:

> There's little point in thinking about the future because, in that sense, one would have to have a model of what the future may hold and therein lies part of the difficulty, that I can't map that future and make any choices. (Participant with PCA)

This impacted how able many dyads felt to effectively plan for the future and many described taking a conscious decision to not think about it, given that there seemed to be little purpose or pleasure in doing so because of the incurable and progressive nature of the diagnosis:

> Every now and again I get down, mainly because if I think too long about what the future holds then… it's counter-productive…It's going to happen. There's nothing you can do about it. You know, it's like one of those things. It's nothing… it's all… you can give a problem a lot of thought if there's an answer; right, do we do this or do we do that? Right, think about it a long time, perhaps worry about it for a couple of days. Right, let's do that. With this, there isn't… What's Plan B? You haven't got a Plan B. And that's this situation. There isn't a Plan B. (Participant with PCA)

In shifting the focus away from an uncertain future, many described their approach as being centred around ideas of 'keeping going' and 'getting on with things', with almost all dyads describing efforts to maintain normality as far as possible (n=17):

> When she [my wife] got her head around what she had, she said, there's nothing I can do about it, we've just got to get on with it. And we just carried on as normal…When it crops up, I deal with it, but 99% of the time, we just carry on as normal…I mean, obviously you gradually get worse and worse, but, you know. (Family carer)

This family carer's comment clearly demonstrates the complexity and significance of temporality in describing the simultaneous day-by-day approach that can exist in combination with broader acknowledgement and anticipation of ongoing decline over the long term.

There were several reports of professionals endorsing or echoing this approach of living in the moment:

> She [doctor] just looked at him and… put her hands on his legs and said, just live your life…just go on and live your life, that's all you can do. (Family carer)

This comment also seemed to address the inevitability of the progression of the disease and the absence of a cure, in describing continuing to 'go on' and to 'live your life' as the only options open to participants in facing the diagnosis. Perhaps it also hints at the lack of published guidance and knowledge about progression of the disease and what to expect that may have enabled or assisted longer term care planning and management. This is corroborated by a recent paper that identified the challenges families and practitioners face in finding tailored, disease-specific information and practical advice about PCA that is evidence based.[67]

Taking a day-by-day approach was preferred and required due to the ever-changing nature of the symptom profile as the disease progressed. Individuals and families were continually responsive, and the majority described being attuned to the necessary ongoing adjustments and adaptations required by the continual change that is characteristic of the disease profile (n=16):

> Yes, as I say, if I let my mind go there [the future], I will probably collapse in a heap, so I find it's best just to deal with things as they present, and just try and think one step ahead, and not too far, because, as I keep being told, every individual with the disease is different, and they can make no… they've got no crystal balls to see into the future, about exactly how it's going to pan [out] for any… one person. (Family carer)

The sense of needing to balance the maintenance of normality in the face of diagnosis-related changes that require ongoing adjustment is also commonly reported throughout the qualitative literature on dementia.[46 61 64 66] A decision to focus on the 'here and now' is also widely reported, owing to the uncertainty or discomfort associated with thoughts about the future, in line with what was reported here.[46 61] These similarities with existing literature perhaps highlight the progressive nature that is common to all variations of AD and dementias more broadly.

## DISCUSSION

This study has illustrated some of the diagnosis-specific stressors associated with PCA and the various ways individuals and families attempt to mediate these. The characteristic visually dominated symptom profile led to primary difficulties in interacting with the physical environment. These were situated within a complex psychosocial environment involving a range of roles and responsibilities requiring reallocation and various individuals' long-standing preferences regarding independence versus asking for, accepting and providing help. The rarity of the condition meant a lack of knowledge and accessible information about the symptoms, disease

course and provision of support for those living with PCA and healthcare professionals, all of which contributed to stress. The temporal context was of particular significance in shaping the stress process in terms of the time of life (eg, employment/retirement status and position in the family), previous levels of performance and engagement in activities/interests, consideration of and concern about the future and the time-limited efficacy of support strategies.

This study offers an original contribution in looking at the day-to-day impacts of progressive visual impairment related to dementia rather than the more typically dominant memory loss.[68] Also, the inductive, qualitative methodology used here offers unique insights to complement the existing PCA literature that is largely laboratory based and concentrated on specifying the cognitive profile and underlying pathology of the disease. This study has gleaned insights firmly grounded in participants' day-to-day experiences, within their home environments and recounted in their own words. This has allowed the full range of experience to be reported and documented. For example, families were able to share the problematic nature of symptoms and ongoing decline and also their resilience in their collaborative and creative approaches to developing coping strategies and in continually adapting to the diagnosis-related changes.

Although this is the first qualitative study of PCA, the findings here bear relevance to existing literature in several ways. The stress caused by the uncertainty and atypical nature of the diagnosis and the impact of lacking disease-specific support and guidance echoes that which other authors have described in relation to other rare diseases.[69] The findings here may also bear relevance for those living with or seeking diagnoses of other rarer types of dementias, into which research is rightfully ever-increasing (eg, behavioural variant frontotemporal dementia and the primary progressive aphasias[70]).

Overall, we need to remain critical of the questions we ask and sensitive to the individual differences in disease profile across and within diagnostic groups, especially in terms of what this means in terms of psychosocial impact for the individual and family. Perhaps the most significant contribution of this paper is in outlining the different mechanisms (eg, visual vs memory problems) that can underpin difficulties with daily activities, even if the psychosocial ramifications of such difficulties are similar or overlapping. Understanding the ways in which the experiences of those with different diagnoses and at different stages of their disease overlap and diverge will be essential if we are to build a knowledge base in which all the complex stories of living with the dementias are told.

A strength of the study is that those with a diagnosis of PCA and a family member were interviewed together and separately, ensuring that multiple perspectives were represented in the data set. In this case, and in contrast to some existing literature on spousal couples living with tAD,[60–62] the relatively well preserved insight of those with PCA resulted in largely congruent accounts of both parties in terms of levels of abilities and shared understandings of the illness experience. However, the discrepancies discussed above—and the varying plausible reasons for them—highlight the importance of interviewing in such a way that rich data from a range of perspectives can be gleaned and also the sensitivities and ethics around the assigning of credit to and/or interpretation of such data. That interviews were conducted in the home environment enriched the data in permitting researchers an in-depth understanding of the everyday physical environment in which difficulties emerged and were responded to and often acted as a useful prompt for participants when discussing challenges and strategies. Working within a process model encouraged consideration of related underlying mechanisms, resulting stressors and responsive coping strategies. The study also makes a broader contribution in highlighting a potential limitation of the Stress Process Model in not taking account of the physical environment as a potential source or mediator of stress, despite suggestions that this may play a particularly significant role for people with PCA[71] and dementia in general.[72–74] A possible limitation of the study is that the home-based nature of the interviews may have deterred dyads who were not managing well from taking part and, as such, the findings may not capture the full range of coping responses to the stress process. The interviews took place at one time point, and the emergent importance of the temporal context may make this another limitation of the study.

In light of these findings, implications for clinicians centre around the need for increased knowledge and provision of information—particularly in a diagnostic context—which is particular to the challenges associated with dementia-related visual impairment and sensitive to the psychosocial ramifications of these difficulties. In addition, dominant difficulties interacting with the physical world may make those with PCA particularly suitable for psychosocial interventions targeted at the marital or family unit as a whole, owing to the relative cognitive strengths of those with PCA (eg, insight, memory and language). Unanswered questions remain about how PCA progresses beyond the moderate stages and also how visual problems may affect people with more typical, memory-led forms of dementia, perhaps at a later stage when they are less easily communicated. Equally, future focused work that factors in the impacts of secondary impairments such as the concurrent memory and language impairments seen in subgroups of our sample here would be helpful to further unpick and illustrate the complexity of the PCA syndrome and the varying impacts it can have. Future research that looks at this both over time and taking account of the multiple perspectives inherent in any dementia journey would constitute valuable and original contributions to knowledge.

## CONCLUSION

This study provides new insights into the stress process for individuals and families living with PCA, from the search for a diagnosis through to the daily challenges of living with dementia-related visual impairment. Increased availability and accessibility of information about PCA, its early symptoms and progression for both healthcare professionals and affected families would be beneficial in aiding timely diagnosis and minimising ongoing stress and uncertainty. Key considerations in the design of supportive interventions for those with PCA would be timeliness and sensitivity to the complexities of the surrounding psychosocial environment within which they must be adopted and adapted to over time.

**Acknowledgements**  We would like to thank all those who participated in this study. This work was supported by Economic and Social Research Council (ESRC)-National Institute for Health Research (NIHR) (UK) grant ES/L001810/1. Grant title: seeing what they see: compensating for cortical visual dysfunction in Alzheimer's Disease. SJC is also supported by the NIHR Queen Square Dementia Biomedical Research Unit and an Alzheimer's Research UK Senior Research Fellowship. RW is supported on a Brunel University London doctoral scholarship.

**Contributors**  EH contributed to study protocol development; data collection, analysis and interpretation; and manuscript preparation (drafting and incorporation of comments/amendments). MPS contributed to study protocol development; data collection, analysis and interpretation; and provided comments on draft manuscripts and approved final manuscript. RW contributed to study protocol development; data collection; and provided comments on draft manuscript. KXXY contributed to study protocol development; recruitment; data collection, analysis and interpretation; and provided comments on draft manuscript. AM contributed to study protocol development, and data collection, analysis and interpretation. MLG contributed to study protocol development and provided comments on draft manuscript. KJG contributed to study protocol development and comments on draft manuscript. SJC contributed to study protocol development and provided comments on draft manuscripts and approved final manuscript.

**Funding**  This study was funded by Economic and Social Research Council and National Institute for Health Research (ES/L001810/1).

**Competing interests**  None declared.

**Patient consent**  Detail has been removed from this case description/these case descriptions to ensure anonymity. The editors and reviewers have seen the detailed information available and are satisfied that the information backs up the case the authors are making.

**Ethics approval**  The study was approved by the National Research Ethics Service Committee – London Queen Square.

**Provenance and peer review**  Not commissioned; externally peer reviewed.

**Data sharing statement**  Data will be made available in accordance with funder guidelines after the completion of the project (March 2018).

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
