## [Reviewer comments · BMJ Open]

ARTICLE DETAILS

TITLE (PROVISIONAL)	'Because my brain isn't as active as it should be, my eyes don't always see' – a qualitative exploration of the stress process for those living with posterior cortical atrophy
AUTHORS	Harding, Emma; Sullivan, Mary Pat; Woodbridge, Rachel; Yong, Keir; McIntyre, Anne; Gilhooly, Mary; Gilhooly, KJ; Crutch, Sebastian

VERSION 1 – REVIEW

REVIEWER	Dr Samrah Ahmed University of Oxford England, UK Research programme in clinical phenotyping in PCA.
REVIEW RETURNED	01-Aug-2017

GENERAL COMMENTS	Harding et al. present a novel approach to interrogating the stressors experienced by PCA patients and, importantly, the impact on their everyday lives. The paper is a collection of rich information, acquired through interviews with patients and their carers. I have several suggestions that may help to present the data in a more accessible way: 1. Reference is made to "existing literature" on stress and coping in dementia, from which the framework for analysis has been drawn. Further information would be useful to detail this literature review.2. The authors make reference to preserved memory as being central to accurate recall in their interviews, yet the neuropsychological assessment shows a subgroup of patients did have memory impairment at interview. It might be useful to stratify the main group into subgroups and examine differences in stressors this way.3. The authors have provided sample sizes to illustrate certain points made. I think this is a very nice approach as it gives an idea of common experiences in what can be a variable patient group. I would propose providing more sample sizes for some of the major points being made, and provided more discussion around the findings that are variable from the typical signal being reported.4. The authors have rich data acquired from carers and patients, although this does not come through in the description of the data. Perhaps the results could be re-organised to show carer and patient perspectives? Or more comment provided to describe similarities/discrepancies between patients and carers?
---

REVIEWER	Professor Peter K Panegyres Neurodegenerative Disorders Research Pty Ltd 4 Lawrence Avenue West Perth WA 6005 Australia
REVIEW RETURNED	11-Sep-2017

GENERAL COMMENTS	An excellent paper and must be published as it emphasizes the psychosocial stressors of patients with posterior cortical atrophy.
---

REVIEWER	Dr Andrea Mayrhofer University of Hertfordshire, England
REVIEW RETURNED	24-Oct-2017

GENERAL COMMENTS	This paper makes an important contribution to understanding the illness trajectory of a sub-type of dementia that frequently occurs in younger onset dementia (YOD). It highlights the specific symptomology that presents in PCA and guides service design for this patient group and their caregivers, particularly in relation to working with residual abilities. The methodological approach employed in this study was described in detail. The rigor involved is evident, and this is reflected in the score sheet. One small correction: On page 12, line 17, please remove the interviewee's name. I look forward to reading this paper in print. With best wishes for continued success, Reviewer
---

VERSION 1 – AUTHOR RESPONSE

Reviewer: 1

Reviewer Name: Dr Samrah Ahmed

Institution and Country: University of Oxford, England, UK

Please state any competing interests or state 'None declared': Research programme in clinical phenotyping in PCA.

Please leave your comments for the authors below

Comment: Harding et al. present a novel approach to interrogating the stressors experienced by PCA patients and, importantly, the impact on their everyday lives. The paper is a collection of rich information, acquired through interviews with patients and their carers. I have several suggestions that may help to present the data in a more accessible way:

Response: Many thanks for your very helpful, considered and encouraging comments on this manuscript. Please find detailed responses to each comment below and corresponding revisions throughout the manuscript. We consider these to have significantly strengthened the paper so thank you once again and we hope you will agree.

1. Reference is made to "existing literature" on stress and coping in dementia, from which the framework for analysis has been drawn. Further information would be useful to detail this literature review.

Thank you for this comment – it is encouraging to know that the readership would value more discussion of the theoretical underpinnings of the paper and the background literature. Please find an additional paragraph with more information about and discussion of the reviewed literature and justification for our analytic framework in the context of that on page 4.

2. The authors make reference to preserved memory as being central to accurate recall in their interviews, yet the neuropsychological assessment shows a subgroup of patients did have memory impairment at interview. It might be useful to stratify the main group into subgroups and examine differences in stressors this way.

Thank you for this important point and of course you are absolutely right that about half of our sample displayed some memory impairment at the time of neuropsychological assessment. Our PCA sample was matched to a comparative typical Alzheimer's disease group (tAD; to be reported elsewhere) and as such varied in terms of disease severity, and with this came varying levels of secondary impairments including memory and language difficulties. In the first instance, we have added an extra limitation in the 'Strengths and Limitations' box addressing this on page 3.

However on broader reflection, we would suggest that we have perhaps presented too 'black and white' a view of PCA throughout the manuscript, focusing possibly too heavily on the core visual processing deficits alone. We had put these at centre stage as the defining feature of PCA as these deficits were common to the majority of our sample and a more unifying feature of their experience. This was in keeping with our methodological rationale – we conducted a thematic analysis, as we were looking for patterns and themes which applied to the whole or majority of the group (i.e. the themes identified would not only bear relevance to those with PCA without memory problems but would have been common to most regardless of the level of memory impairment). Having said this as discussed in comment (3) below, there is of course a wealth of variation in individual experience and differing levels of memory impairment may indeed have contributed to some deviations from the 'typical signal' being reported. We hope we have addressed parts of this comment and comment (3) by providing examples of some of these variations – particularly relating to the impact of concurrent memory impairment – on pages 12 and 15 in the revised manuscript.

Further, whilst we certainly agree that it would be interesting and clinically relevant to stratify a PCA sample by presence/absence or level of memory impairment, we would argue that that would be beyond the scope of this current paper which – as the first qualitative exploration of PCA – necessarily takes a very general and broad approach. As reported, we did interview a comparative sample of dyads in which one had a diagnosis of tAD, but the volume of data across the two groups was too large for thorough presentation in a single paper. To stratify our findings fully according to memory impairment would require us to return to the raw data and run two separate thematic analyses for the groups (PCA +/- memory impairment) in order to make comparisons. Our concern with doing this would be that with as broad a focus as 'stress and coping' the results of a stratified analysis would far exceed 10,000 words and in turn be less accessible and/or desirable for publication. We would however certainly be keen to explore differences in particular categories of stressors in stratified subgroups of PCA in future work (e.g. the uptake of physical aids and adaptations by those with and without memory impairment; the impact on couple-hood for those with and without language impairment) in order to add a depth of understanding of the varied clinical presentations of PCA to the breadth we have attempted to cover in this initial exploration. To acknowledge this we have added a sentence to the Discussion (page 20) about the benefits of taking a stratified approach to the PCA syndrome in future work.

Overall, we are in total agreement that a more nuanced presentation of PCA is required (however broad the focus of the paper!), and hope we have strengthened the manuscript by addressing this early in the Introduction (page 3). We have replaced the section which referred to a distinction between dominant visual processing problems and relatively preserved episodic memory with one which acknowledges that while visual processing problems will dominate, these may be accompanied by secondary symptoms for a proportion of people with PCA, with reference to your work outlining the memory impairments that can be evident for PCA patients even at initial clinical presentation, and Crutch et al.'s work on early language difficulties that some people with PCA demonstrate. More generally, we have adjusted our phrasing throughout (e.g. adding terms such as 'most of', 'relative', 'typically') in an attempt to build a more nuanced picture of the heterogeneity in the observed patterns of deficits in the PCA symptom profile.

Lastly, our methodological decision to interview dyads together and separately was in part because of the anticipation of potential memory problems and/or word-finding difficulties in both groups, though on reflection this was not made clear in the manuscript and we have added a revision to address this on page 6. It is perhaps worth mentioning that as discussed in the findings, (particularly pages 14-16) we saw relatively little discrepancy and need for 'filling in' by carers of the PCA group (especially in comparison to the tAD group to be reported elsewhere) but there certainly were instances of discrepancies in accounts and we hope to have addressed any concerns related to these under comment (4) below.

3. The authors have provided sample sizes to illustrate certain points made. I think this is a very nice approach as it gives an idea of common experiences in what can be a variable patient group. I would propose providing more sample sizes for some of the major points being made, and provided more discussion around the findings that are variable from the typical signal being reported.

Response: Thank you also for this helpful comment and it is encouraging to know that the sample size indications help to make the paper more accessible and also that descriptions of some variation from these central patterns are useful regarding the heterogeneity of the sample. We hope we have addressed the first part of this comment by providing additional sample sizes throughout the revised manuscript (pages 9, 12, 17 and 18). We have also attempted to provide some rich examples of divergence relating to: the efficiency of the diagnostic process (page 9), the variation in marital history and the impact on stress/coping (page 13), opting not to adopt a 'teamwork' approach (page 14) and the impact of the less common memory impairments as described in the response to comment (2) above (pages 12 and 15).

4. The authors have rich data acquired from carers and patients, although this does not come through in the description of the data. Perhaps the results could be re-organised to show carer and patient perspectives? Or more comment provided to describe similarities/discrepancies between patients and carers?

Response: Many thanks for this comment also. It was certainly our intention to glean rich data from both perspectives and this informed our methodological decisions regarding the format, structure and order of the interviews we conducted and it is helpful to know that this was not fully communicated. As discussed, we certainly observed more agreement and overlap between patient and carer data in the PCA group and so would have reservations about re-organising the data into their respective perspectives in case this (a) alluded to or created an impression of more discrepancy or diversion than was witnessed or characteristic of this group and (b) if this resulted in some seemingly 'redundant' sections in which limited differences could be reported.

Having said that, this general consensus/agreement is probably largely due to the breadth of the issues reported and discussed here, and this is absolutely not to dispute that a detailed exploration comparing the particular variations in experiences of the person taking a caring versus patient role in the disease course of PCA would be interesting, important and valid, and this is something we would hope to build on the current study findings with in the future. Within this initial, exploratory paper we sought more to acknowledge that there are multiple perspectives within any one illness journey and we would certainly hope to unpick the specificities of those in more detail at a later date, with greater parameters and more depth (as with the stratification according to secondary impairments). Having said that, we hope to have addressed the second part of this comment, in providing more description around the similarities and discrepancies between accounts on pages 14-16 of the Findings section and page 20 of the Discussion.

Reviewer: 2

Reviewer Name: Professor Peter K Panegyres

Institution and Country: Neurodegenerative Disorders Research Pty Ltd, 4 Lawrence Avenue, West Perth WA 6005, Australia

Please state any competing interests or state 'None declared': None declared

Please leave your comments for the authors below

An excellent paper and must be published as it emphasizes the psychosocial stressors of patients with posterior cortical atrophy.

Very many thanks indeed for your positive and encouraging comments on our work!

Reviewer: 3

Reviewer Name: Dr Andrea Mayrhofer

Institution and Country: University of Hertfordshire, England

Please state any competing interests or state 'None declared': None declared

Please leave your comments for the authors below

Comment: This paper makes an important contribution to understanding the illness trajectory of a sub-type of dementia that frequently occurs in younger onset dementia (YOD).

It highlights the specific symptomology that presents in PCA and guides service design for this patient group and their caregivers, particularly in relation to working with residual abilities.

The methodological approach employed in this study was described in detail. The rigour involved is evident, and this is reflected in the score sheet.

One small correction: On page 12, line 17, please remove the interviewee's name.

I look forward to reading this paper in print.

With best wishes for continued success

Response: Many thanks for your encouraging and thoughtful comments. We are delighted that you found the implications for service design and importance of harnessing residual abilities to be clearly presented, and also that we have effectively demonstrated the rigour in our approach. Many thanks for pointing out the pseudonym on page 12, this has now been replaced with [He].